# Temporally Detailed Hypergraph Neural ODEs for Disease Progression Modeling

**Tingsong Xiao**[1]     **Yao An Lee**[2,4]     **Zelin Xu**[1]     **Yupu Zhang**[1]     **Zibo Liu**[1]
**Yu Huang**[2,3]     **Jiang Bian**[2,3]     **Jingchuan Guo**[2,4]     **Zhe Jiang**[1]

[1]Department of Computer and Information Science and Engineering, University of Florida
[2]Regenstrief Institute
[3]Department of Biostatistics and Health Data Science, Indiana University
[4]Department of Pharmacy Practice, Purdue University
{xiaotingsong, zhe.jiang}@ufl.edu

## Abstract

Disease progression modeling aims to characterize and predict how a patient's disease complications worsen over time based on longitudinal electronic health records (EHRs). For diseases such as type 2 diabetes, accurate progression modeling can enhance patient sub-phenotyping and inform effective and timely interventions. However, the problem is challenging due to the need to learn continuous-time progression dynamics from irregularly sampled clinical events amid patient heterogeneity (*e.g.,* different progression rates and pathways). Existing mechanistic and data-driven methods either lack adaptability to learn from real-world data or fail to capture complex continuous-time dynamics on progression trajectories. To address these limitations, we propose **T**emporally **D**etailed **H**ypergraph **N**eural **O**rdinary **D**ifferential **E**quation (**TD-HNODE**), which represents disease progression on clinically recognized trajectories as a temporally detailed hypergraph and learns the continuous-time progression dynamics via a neural ODE framework. TD-HNODE contains a learnable TD-Hypergraph Laplacian that captures the interdependency of disease complication markers within both intra- and inter-progression trajectories. Experiments on two real-world clinical datasets demonstrate that TD-HNODE outperforms multiple baselines in modeling the progression of type 2 diabetes and related cardiovascular diseases.

## 1 Introduction

Many chronic diseases follow a progressive trajectory with multiple complications that worsen over time (Uddin et al., 2023). For example, one patient with type 2 diabetes may gradually develop retinopathy, which advances to visual impairment and eventually to blindness. Another patient may develop hypertension, then atrial fibrillation, and ultimately heart failure. Other patients may experience complications on both trajectories. The goal of disease progression modeling is to learn how a patient's complications evolve over time from longitudinal electronic health records (EHRs) (Mould, 2012; Wang et al., 2014; Xiao et al., 2025a). Specifically, each clinical visit is associated with a risk factor feature vector (*e.g.,* lab results, medications) and a complication marker vector (*e.g.,* presence of hypertension, atrial fibrillation, or cerebrovascular disease). The task is to predict the complication marker vector at the next visit, with the constraint that marker progression follows a set of clinically verified trajectories. Disease progression modeling plays a crucial role in patient sub-phenotyping (*i.e.,* grouping patients into categories based on heterogeneous progression patterns) and informing effective and timely treatment (Buil-Bruna et al., 2015; Prague et al., 2013).

However, the problem poses several technical challenges. First, patient records of hospital visits are often sampled at irregular time points, although the underlying disease conditions evolve in continuous time. Second, many chronic and progressive diseases—such as type 2 diabetes, Alzheimer's disease, chronic kidney disease (CKD), cancer (*e.g.,* breast or prostate), and cardiovascular diseases—have clinically recognized progression trajectories that are routinely used to guide treatment planning. Incorporating these validated pathways into a data-driven model is non-trivial, as it requires a structured representation that can encode multi-step, high-order progression dependencies

beyond pairwise relations. Third, the progression dynamics are heterogeneous among patients, as reflected by the varying progression rates and progression trajectories (*e.g.,* some patients rapidly develop kidney damage while others remain stable for many years, and some patients may experience neuropathy followed by a foot ulcer).

Existing works on disease progression modeling can be broadly categorized into mechanistic and data-driven approaches (Cook and Bies, 2016; Mould, 2012). Mechanistic models (van Schaick et al., 2015; Shahar, 1995) incorporate biological, pathophysiological, and pharmacological processes into the modeling structure, providing enhanced interpretability but with limited adaptability to real-world data. Data-driven approaches can be further divided into traditional machine learning and deep learning. Traditional machine learning methods often use hidden Markov models (Jackson et al., 2003; Sukkar et al., 2012; Liu et al., 2015) to capture transition probabilities between disease stages, but they typically rely on strong assumptions about data distribution at known progression stages and cannot learn implicit stages based on complex feature representations. As large-scale EHRs have become increasingly available over the last decade, deep learning methods have been widely developed (Shickel et al., 2017; Solares et al., 2020), including recurrent neural networks (*e.g.,* LSTM) (Zhang, 2019; Zhang et al., 2019; Sohn et al., 2020), attention-based models (*e.g.,* Transformers) (Zhang, 2019; Zisser and Aran, 2024; Xiao et al., 2025b; Shmatko et al., 2025; Song et al., 2025), and Neural Ordinary Differential Equations (Neural ODEs) (Chen et al., 2018; Goyal and Benner, 2023; Chen et al., 2024a). Neural ODE models can capture the continuous-time dynamics of disease progression based on irregular-time clinical events, but existing models (Qian et al., 2021; Dang et al., 2023) fail to incorporate the clinically verified progression pathways. Continuous-time graph neural networks (Rossi et al., 2020; Tian et al., 2021; Liu et al., 2024; Cheng et al., 2024) can potentially represent known progression trajectories as a directed graph, where nodes are complication markers and temporal edges represent progression between them, but a normal graph only captures pairwise interactions (between one complication and its immediate predecessor or successor) and thus misses high-order interactions across all complication nodes along a trajectory (pathway) (Yoon et al., 2020).

To address these limitations, we propose the **T**emporally **D**etailed **H**ypergraph **N**eural **O**rdinary **D**ifferential **E**quation (**TD-HNODE**), which represents disease progression on clinically recognized trajectories as a temporally detailed hypergraph and learns the continuous-time progression dynamics via a neural ODE framework. The key idea is as follows. We adopt a Neural ODE to model continuous-time disease progression from irregular-time patient records, where the temporal gradient of disease states is governed by a hypergraph Laplacian that captures high-order interactions among complication markers along clinically known progression trajectories. A standard hypergraph Laplacian, however, is static and fails to reflect patient-specific temporal dynamics. To this end, TD-HNODE introduces a *learnable TD-Hypergraph Laplacian* that contains two novel components: (1) an attention-based incidence matrix that assigns time-aware, patient-specific importance to each marker within a trajectory (*i.e.,* intra-trajectory dynamics), and (2) a learnable hyperedge weight matrix that captures dependencies across different trajectories (*i.e.,* inter-trajectory correlations). It is worth noting that our proposed framework differs from existing temporal hypergraph neural networks (Lee and Shin, 2023; Liu et al., 2022; Lee and Shin, 2021), which assign timestamps at the level of entire hyperedges, failing to capture the fine-grained temporal progression details within each hyperedge. Although some methods adopt recurrent modules (*e.g.,* RNNs (Wang et al., 2024a; Younis and Ahmadi, 2024)) and others incorporate continuous-time dynamics (*e.g.,* Neural ODEs (Yao et al., 2023; Luo et al., 2023)), they still construct hypergraph snapshots over discrete intervals, where hyperedges remain static units without modeling marker-level timestamps. We validate TD-HNODE on two real-world EHR datasets, showing consistent improvements over baselines in modeling the progression of type 2 diabetes and cardiovascular diseases.

## 2 PROBLEM STATEMENT

### 2.1 PRELIMINARIES

A patient's medical history can be represented as a sequence of hospital visits called **encounters**. We denote an encounter for patient $u$ at time $t_k$ as $\{\mathbf{x}_u(t_k); \mathbf{y}_u(t_k)\}$, where $k$ denotes the index of the $k$-th encounter (timestamp) in the patient's encounter sequence, and $\mathbf{x}_u(t_k) \in \mathbb{R}^c$ is a vector of **risk factors** ($c$ is the number of risk factors), such as medications, laboratory test results, and

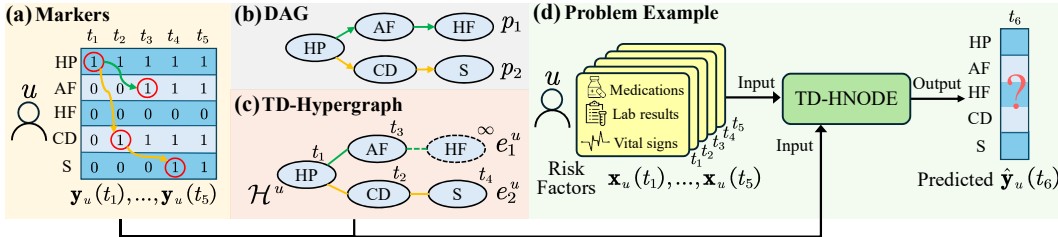

Figure 1: (a) Patient $u$'s marker status vector from $t_1$ to $t_5$; (b) The corresponding DAG of trajectories $p_1$ and $p_2$; (c) Patient $u$'s TD-Hypergraph with two temporally detailed hyperedges $e_1^u, e_2^u$; (d) An example of problem definition showing TD-HNODE's input and output.

vital signs, and $\mathbf{y}_u(t_k) \in \{0,1\}^n$ is a vector of **disease complication markers** (1 for presence, 0 for absence), such as hypertension (HP), atrial fibrillation (AF), heart failure (HF), cerebrovascular disease (CD), and stroke (S). The set of all $n$ markers is denoted as $\mathcal{V} = \{v_i \mid i = 1, \ldots, n\}$.

The marker vector $\mathbf{y}_u(t)$ evolves over time in the encounter sequence of patient $u$, reflecting the progression of disease states. How the markers evolve follows a set of **disease progression trajectories** (or **pathways**) that can be constructed based on clinical knowledge. Formally, a trajectory is defined as an ordered sequence of distinct markers, denoted as $p_j = <v_1^j, v_2^j, \ldots, v_{|p_j|}^j>$, where $v_i^j \in \mathcal{V}$ represents the $i$-th marker in the $j$-th trajectory, and $|p_j|$ is the number of markers.

Figure 1(a) provides an example of the temporal records of a patient's five disease complication markers (HP, AF, HF, CD, and S) from $t_1$ to $t_5$. Figure 1(b) shows two clinically known progression trajectories (in green and yellow) that these markers follow, *i.e.,* $p_1 = <HP, AF, HF>$ and $p_2 = <HP, CD, S>$. The fact that the patient's markers follow these trajectories is highlighted by the red circles as well as the green and yellow arrows in Figure 1(a). For instance, once the patient has hypertension ($HP = 1$) at time $t_1$, the status will persist. The patient's disease progresses from hypertension to cerebrovascular disease ($CD = 1$) at time $t_2$ and to stroke ($S = 1$) at $t_4$, following the yellow trajectory. Similarly, the disease progresses from hypertension to atrial fibrillation at $t_3$ ($AF = 1$) following the green trajectory. Note that the patient's disease complication markers have not yet progressed to heart failure (HF) by $t_5$, but it could happen at a future time.

We assume that the disease markers in the trajectory are **irreversible**, *i.e.,* the binary status of markers can only transit from 0 to 1 (or stay the same) within an encounter sequence. This assumption is reasonable for many chronic diseases—such as type 2 diabetes, Alzheimer's disease, and chronic kidney disease—that exhibit irreversible progression (*e.g.,* organ fibrosis, vascular damage) (Wu et al., 2021; Kazemian et al., 2019; Bhatwadekar et al., 2021; Wang et al., 2024b).[1]

The set of clinically known progression trajectories forms a Directed Acyclic Graph (DAG), as illustrated in Figure 1(b). To capture the higher-order marker patterns within trajectories, we propose to represent trajectories using a hypergraph (Gallo et al., 1993). Specifically, we represent the set of markers within the same trajectory as a hyperedge. There are two key advantages: (1) a hyperedge can connect multiple markers in an entire trajectory (pathway), enabling the modeling of high-order dependencies beyond pairwise relations (Feng et al., 2019; Gao et al., 2022); and (2) different hyperedges overlap with each other through common markers (potentially pivotal nodes) (Chitra and Raphael, 2019), making it easier to model interdependency across progression pathways.

We formally define the **disease progression hypergraph** as $\mathcal{H} = (\mathcal{V}, \mathcal{E})$, where $\mathcal{V}$ is the set of **markers**, also referred to as **nodes** throughout the paper, and $\mathcal{E}$ denotes the set of hyperedges, each corresponding to a predefined trajectory (we use the terms 'hyperedge', 'pathway', and 'trajectory' interchangeably). Specifically, $\mathcal{E} = \{e_1, e_2, \ldots, e_m\}$, and $e_j = \{v_1^j, v_2^j, \ldots, v_{|e_j|}^j\}$, where $m$ is the total number of predefined trajectories, and $|e_j|$ is the number of markers in the $j$-th trajectory.

---

[1]For lab-derived markers such as "HbA1c High/Low" and "Poor Lipid/BP," we use their *first occurrence* as a progression signal in $\mathbf{y}$, indicating that the patient's chronic stage has reached a clinically significant level (*e.g.,* once a patient has experienced "Poor Lipid," it signals progression to that stage regardless of later fluctuations). This design choice was confirmed by a clinician (M.D.) among our co-authors. Meanwhile, the raw fluctuating lab values (*e.g.,* GFR, HDL, Triglycerides) are incorporated through the risk factor features $\mathbf{x}$.

To capture each patient's actual progression, we define a **Temporally Detailed Hypergraph (TD-Hypergraph)** $\mathcal{H}^u = (\mathcal{V}, \mathcal{E}^u)$, where $\mathcal{E}^u = \{e_1^u, e_2^u, \ldots, e_m^u\}$ is a set of **temporally detailed hyperedges**. Each hyperedge records both the markers and the timestamps at which they were first observed for patient $u$: $e_j^u = \{(v_1^j, t_1), (v_2^j, t_2), \ldots, (v_k^j, t_k), (v_{k+1}^j, \infty), \ldots, (v_{|e_j|}^j, \infty)\}$, where markers up to $v_k^j$ have been observed (with $t_i$ being the timestamp of first occurrence), and the placeholder $\infty$ indicates markers not yet observed. Since a patient may not develop all markers in a predefined trajectory, the observed portion can be shorter than the full trajectory. For example, as shown in Figure 1(a), the patient's observed progression up to $t_5$ yields two temporally detailed trajectories: $p_1^u = < (HP, t_1), (AF, t_3) >$ (shorter than the full trajectory $p_1 = < HP, AF, HF >$) and $p_2^u = < (HP, t_1), (CD, t_2), (S, t_4) >$. Figure 1(c) shows the resulting TD-Hypergraph at time $t_5$: the markers have progressed to $S$ (stroke) but not $HF$ (heart failure) yet. The TD-Hypergraph is time-dependent—it evolves as complication markers progress over time, reflected by timestamp (temporal details) updates on hyperedges.

A list of commonly used notations is provided in Table 3 (Appendix A).

## 2.2 PROBLEM DEFINITION

Given patient encounter data with risk factors $\{\mathbf{x}_u(t_1), \ldots, \mathbf{x}_u(t_k)\}_{u=1}^N$, target complication markers $\{\mathbf{y}_u(t_1), \ldots, \mathbf{y}_u(t_k); \mathbf{y}_u(t_{k+1})\}_{u=1}^N$, and the TD-Hypergraph $\{\mathcal{H}^u\}_{u=1}^N$, the problem aims to learn a TD-HNODE model $f_{\boldsymbol{\Theta}}$ such that:

$$\hat{\mathbf{y}}_u(t_{k+1}) = f_{\boldsymbol{\Theta}}\left(\mathbf{x}_u(t_1), \ldots, \mathbf{x}_u(t_k), \ \mathbf{y}_u(t_1), \ldots, \mathbf{y}_u(t_k); \ \mathcal{H}^u\right). \tag{1}$$

The objective is to minimize the binary cross-entropy loss:

$$\min_{\boldsymbol{\Theta}} \frac{1}{N} \sum_{u=1}^N \mathcal{L}\left(\hat{\mathbf{y}}_u(t_{k+1}), \ \mathbf{y}_u(t_{k+1})\right). \tag{2}$$

Figure 1(d) provides an illustrative example. The TD-HNODE model takes as input patient $u$'s risk factors $\mathbf{x}_u(t_1), \ldots, \mathbf{x}_u(t_5)$ (*e.g.,* medications, lab results, and vital signs, Figure 1(d) left), marker status $\mathbf{y}_u(t_1), \ldots, \mathbf{y}_u(t_5)$ (Figure 1(a)), and a TD-Hypergraph $\mathcal{H}^u$ (Figure 1(c)). The model predicts future marker status at $t_6$ (Figure 1(d) right).

## 3 METHODOLOGY

**An overview of our model framework:** As shown in Figure 2(a), TD-HNODE employs Neural ODE to model continuous-time disease progression from irregular-time encounters, where nodes represent complication markers and hyperedges represent clinically known progression trajectories (patient index $u$ omitted for simplicity). At each encounter time $t_k$, risk factors $\mathbf{x}(t_k)$ and complication markers $\mathbf{y}(t_k)$ are embedded into node representations of the TD-Hypergraph $\mathcal{H}^u$, which are used to construct a learnable TD-Hypergraph Laplacian $\tilde{\mathbf{L}}(t)$ (Figure 2(b)) capturing intra-trajectory dynamics and inter-trajectory dependencies. Let $\mathbf{S}(t) \in \mathbb{R}^{n \times d}$ denote the hidden state of markers at time $t$, where $n$ is the number of markers and $d$ is the embedding dimension. The Laplacian $\tilde{\mathbf{L}}(t)$, together with risk factors and hidden state $\mathbf{S}(t_k)$, is passed to the Neural ODE solver to update the latent state $\mathbf{S}(t_{k+1})$, which is then decoded to predict marker status $\hat{\mathbf{y}}(t_{k+1})$.

### 3.1 NEURAL ODE AND HYPERGRAPH NEURAL NETWORK

**General Neural ODE.** A Neural ODE learns a continuous dynamic function $\mathbf{S}(t)$:

$$\frac{d\mathbf{S}(t)}{dt} = f\left(t, \mathbf{S}(t), \mathbf{x}(t); \boldsymbol{\Theta}\right), \tag{3}$$

where $\boldsymbol{\Theta}$ are learnable parameters and $f(\cdot)$ models the temporal gradient of the disease state. The hidden state is initialized as $\mathbf{S}(t_1) = \mathbf{0} \in \mathbb{R}^{n \times d}$, with the patient's initial conditions introduced through the risk factor embedding $h(\mathbf{x}(t_1))$ in Eq. (4).

**Static hypergraph Laplacian.** We instantiated $f(\cdot)$ using a hypergraph Laplacian $\mathbf{L}$ (Feng et al., 2019) to enable multi-way message passing over disease trajectories:

$$f\left(t, \mathbf{S}(t), \mathbf{x}(t); \boldsymbol{\Theta}\right) = -\mathbf{L}\left[\mathbf{S}(t) + h(\mathbf{x}(t))\right]\boldsymbol{\Theta}, \tag{4}$$

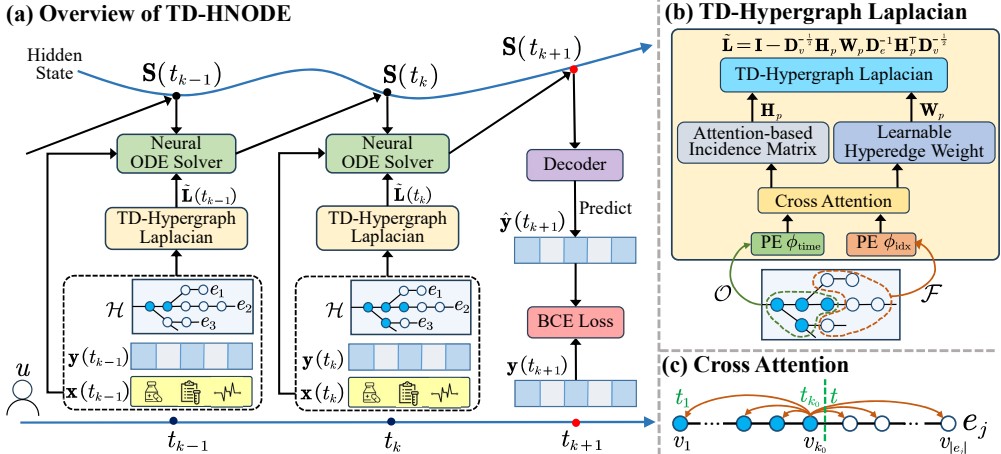

Figure 2: (a) Overview of the TD-HNODE, with an example of patient $u$'s encounter sequence; (b) The TD-Hypergraph Laplacian module combining an Attention-based Incidence Matrix and a Learnable Hyperedge Weight Matrix; (c) An illustration of cross attention within hyperedge $e_j$.

where $h(\cdot)$ maps $\mathbf{x}(t)$ into the same space as $\mathbf{S}(t)$, and $\boldsymbol{\Theta} \in \mathbb{R}^{d \times d}$ is a learnable transformation matrix. The negative sign simulates diffusion-like propagation among markers (Ji et al., 2022). The common hypergraph Laplacian $\mathbf{L}$ is defined as:

$$\mathbf{L} = \mathbf{I} - \mathbf{D}_v^{-1/2} \mathbf{H} \mathbf{W} \mathbf{D}_e^{-1} \mathbf{H}^\top \mathbf{D}_v^{-1/2}, \tag{5}$$

Intuitively, this Laplacian governs how information diffuses among markers: markers connected by the same hyperedge (trajectory) exchange signals, with normalization ensuring balanced influence regardless of node or hyperedge size. Here, $\mathbf{I} \in \mathbb{R}^{n \times n}$ is an identity matrix, $\mathbf{H} \in \{0,1\}^{n \times m}$ is the incidence matrix with $\mathbf{H}(i,j) = 1$ if marker $v_i$ belongs to hyperedge $e_j$, and $\mathbf{H}(i,j) = 0$ otherwise. $\mathbf{W} \in \mathbb{R}^{m \times m}$ is a diagonal hyperedge weight matrix. The node and hyperedge degree matrices are $\mathbf{D}_v(i,i) = \sum_j \mathbf{H}(i,j) \mathbf{W}(j,j)$, and $\mathbf{D}_e(j,j) = \sum_i \mathbf{H}(i,j)$.

**Limitation.** However, the above static Laplacian treats all markers equally within each hyperedge ($\mathbf{H}$ is binary) and uses fixed diagonal weights ($\mathbf{W}$), failing to capture: (1) time-varying marker importance within a trajectory, and (2) inter-trajectory dependencies due to shared markers and correlated temporal dynamics. We next extend $\mathbf{L}$ to a learnable, temporally adaptive $\tilde{\mathbf{L}}(t)$ (Eq. 13).

### 3.2 LEARNABLE TD-HYPERGRAPH LAPLACIAN

We proposed two key enhancements: (1) Attention-based Incidence Matrix: we replaced the binary incidence matrix with a learnable attention mechanism, allowing the model to assign time-aware, patient-specific importance to each marker node within a progression trajectory; (2) Learnable Hyperedge Weights: instead of assigning fixed weights to hyperedges, we introduced learnable weights that captured inter-trajectory dependencies based on shared markers and correlation patterns. This design allowed TD-HNODE to perform high-order message passing guided by clinical knowledge while adapting to patient-specific disease trajectories.

#### 3.2.1 ATTENTION-BASED INCIDENCE MATRIX

To account for temporal dynamics and the varying importance of markers in a trajectory, we designed an adaptive incidence matrix based on cross-attention within each temporally detailed hyperedge. For a given time $t$, we define $t_{k_0}$ as the timestamp of the patient's most recent encounter before $t$, and the model uses all sequential encounter data up to $t_{k_0}$ (*i.e.*, encounters at $t_k$ for $1 \leq k \leq k_0$) to construct the observed progression trajectory and the TD-Hypergraph Laplacian. Given a temporally detailed hyperedge $e_j = \{(v_1^j, t_1), (v_2^j, t_2), \ldots, (v_{k_0}^j, t_{k_0}), (v_{k_0+1}^j, \infty), \ldots, (v_{|e_j|}^j, \infty)\}$, we denoted the current progression point at time $t_{k_0}$ as $v_{k_0}$, omitting the trajectory index $j$ for notational simplicity. Intuitively, the current progression point divides each trajectory into markers the patient has already developed and markers that may develop in the future. Based on this, we split $e_j$ into

two subsets: a *past set* $\mathcal{O}^j = \{v_1, ..., v_{k_0}\}$ containing already observed markers, and a *potential set* $\mathcal{F}^j = \{v_{k_0+1}, ..., v_{|e_j|}\}$ containing markers not yet observed.

Each marker $v_i \in e_j$ had an initial embedding $\mathbf{b}_i \in \mathbb{R}^d$, obtained by applying a learnable multilayer perceptron to the one-hot encoding of marker $v_i$. Since observed markers have real timestamps while future markers do not, we used different positional encodings for each group: continuous-time encoding for observed markers and discrete index-based encoding for future markers. Specifically, $\phi(i) = \phi_{\text{time}}(t_i)$ if $v_i \in \mathcal{O}^j$, and $\phi(i) = \phi_{\text{idx}}(i)$ if $v_i \in \mathcal{F}^j$, where $\phi_{\text{time}}(t_i) \in \mathbb{R}^d$ was a continuous-time encoding (Xu et al., 2020), and $\phi_{\text{idx}}(i) \in \mathbb{R}^d$ was a discrete index-based encoding (Vaswani et al., 2017).

We computed the query vectors as:

$$\mathbf{q}_i^{\text{time}} = (\mathbf{b}_i + \phi_{\text{time}}(t_i))\mathbf{W}_Q \quad \text{if } v_i \in \mathcal{O}^j, \tag{6}$$

$$\mathbf{q}_i^{\text{idx}} = (\mathbf{b}_i + \phi_{\text{idx}}(i))\mathbf{W}_Q \quad \text{if } v_i \in \mathcal{F}^j, \tag{7}$$

where $\mathbf{W}_Q \in \mathbb{R}^{d \times d}$ is a learnable projection matrix. In other words, each query encodes "where the patient currently is" in a trajectory, combining the marker's identity with its temporal or positional context. Key and value vectors were constructed analogously, each using its own learnable projection matrix ($\mathbf{W}_K, \mathbf{W}_V \in \mathbb{R}^{d \times d}$) and the same time/index positional encodings.

We then computed the attention weight from the current progression point $v_{k_0}$ to each other marker $v_i \in e_j$, separately for observed and unobserved markers using their respective encodings. We first define the softmax normalization terms over the past and potential sets:

$$Z_\mathcal{O} = \sum_{l \in \mathcal{O}^j} \exp\big(\mathbf{q}_{k_0}^{\text{time}} \cdot \mathbf{k}_l^{\text{time}}/\sqrt{d}\big), \quad Z_\mathcal{F} = \sum_{l \in \mathcal{F}^j} \exp\big(\mathbf{q}_{k_0}^{\text{idx}} \cdot \mathbf{k}_l^{\text{idx}}/\sqrt{d}\big). \tag{8}$$

The attention weight is then:

$$\alpha_j(i, k_0) = \begin{cases} \exp\big(\mathbf{q}_{k_0}^{\text{time}} \cdot \mathbf{k}_i^{\text{time}}/\sqrt{d}\big) \ / \ Z_\mathcal{O} & \text{if } v_i \in \mathcal{O}^j, \\ \exp\big(\mathbf{q}_{k_0}^{\text{idx}} \cdot \mathbf{k}_i^{\text{idx}}/\sqrt{d}\big) \ / \ Z_\mathcal{F} & \text{if } v_i \in \mathcal{F}^j. \end{cases} \tag{9}$$

Intuitively, $\alpha_j(i, k_0)$ measures how much the current progression point $v_{k_0}$ should attend to marker $v_i$: using time-aware encodings for already-observed markers and position-based encodings for markers that may develop in the future. We then constructed the adaptive incidence matrix $\mathbf{H}_p$ by modulating each entry with the cross-attention weight from the current marker $v_{k_0}$ to the marker $v_i$:

$$\mathbf{H}_p(i, j) = \begin{cases} \mathbf{H}(i, j) \cdot \alpha_j(i, k_0) & \text{if } v_i \in e_j, \\ 0 & \text{otherwise.} \end{cases} \tag{10}$$

Here, $\alpha_j(i, k_0)$ encodes the directional and time-aware importance of marker $v_i$ under the current progression context. This formulation allows the incidence matrix to capture both structural relations and temporal dynamics, reflecting the evolving role of each marker during disease progression.

### 3.2.2 LEARNABLE HYPEREDGE WEIGHT MATRIX

Traditional hypergraph-based methods typically assume a fixed hyperedge weight matrix $\mathbf{W} \in \mathbb{R}^{m \times m}$, where $m$ is the number of hyperedges. However, this assumption fails to capture variable correlation strengths in patient-specific progression across trajectories. To address this, we introduced a learnable hyperedge weight matrix $\mathbf{W}_p \in \mathbb{R}^{m \times m}$ based on hyperedge representation, which modeled dynamic dependencies among trajectories. The key idea was to derive trajectory-level embeddings from their constituent markers and compute trajectory similarity in a learned latent space.

For each marker $v_i \in e_j$, we computed its context-enhanced representation $\tilde{\mathbf{v}}_i$ using self-attention within the subset it belongs to. Specifically, self-attention was performed separately over the past set $\mathcal{O}^j$ and the potential set $\mathcal{F}^j$, producing $\tilde{\mathbf{v}}_i = \text{SelfAttn}(v_i, \mathcal{O}^j)$ if $v_i \in \mathcal{O}^j$, and $\tilde{\mathbf{v}}_i = \text{SelfAttn}(v_i, \mathcal{F}^j)$ if $v_i \in \mathcal{F}^j$. Here, $\text{SelfAttn}(v_i, \cdot)$ denotes standard scaled dot-product attention with $v_i$ as query and all other markers in the same subset as keys and values. The resulting vector $\tilde{\mathbf{v}}_i \in \mathbb{R}^d$ captured

context-specific information and was used to form hyperedge-level representations. Specifically, for hyperedge $e_j \in \mathcal{E}$, we aggregated the value vectors $\tilde{\mathbf{v}}_i$ to obtain a trajectory-level representation:

$$\mathbf{g}_j = \text{Aggregate}\{\tilde{\mathbf{v}}_i \mid v_i \in e_j\}, \tag{11}$$

where $\text{Aggregate}(\cdot)$ is a differentiable pooling function, such as average pooling.

We aggregated the trajectory-level embeddings from all hyperedges $\mathcal{E}$ into trajectory embedding matrix $\mathbf{G} = [...; \mathbf{g}_j; ...] \in \mathbb{R}^{m \times d}$, where the $j$-th row of $\mathbf{G}$, *i.e.,* , $\mathbf{g}_j$, encoded the representation of the $j$-th trajectory (hyperedge). We then projected this matrix into a latent space using a trainable linear transformation: $\tilde{\mathbf{G}} = \mathbf{G}\mathbf{W}_{\mathcal{E}}$, where $\mathbf{W}_{\mathcal{E}} \in \mathbb{R}^{d \times d}$. The latent trajectory embedding matrix $\tilde{\mathbf{G}}$ was then used to compute the trajectory correlation matrix, also called *learnable hyperedge weight matrix*:

$$\mathbf{W}_p = \tilde{\mathbf{G}}\tilde{\mathbf{G}}^{\top} \in \mathbb{R}^{m \times m}. \tag{12}$$

This learnable matrix $\mathbf{W}_p$ captures data-driven similarities between all trajectories, allowing the model to emphasize more relevant progression pathways as well as their interdependency. For example, the retinopathy trajectory and the nephropathy trajectory may receive a high correlation weight in $\mathbf{W}_p$ because they share common early markers and often co-progress in diabetic patients.

We combined the adaptive incidence $\mathbf{H}_p$ and the learnable hyperedge weight matrix $\mathbf{W}_p$ to form our Knowledge-Infused TD-Hypergraph Laplacian:

$$\tilde{\mathbf{L}} = \mathbf{I} - \mathbf{D}_v^{-\frac{1}{2}} \mathbf{H}_p \mathbf{W}_p \mathbf{D}_e^{-1} \mathbf{H}_p^{\top} \mathbf{D}_v^{-\frac{1}{2}}, \tag{13}$$

where $\mathbf{I}$ is the $n \times n$ identity matrix. In this way, $\tilde{\mathbf{L}}$ encodes intra-trajectory time-sensitive marker dependencies (through $\mathbf{H}_p$) and inter-trajectory correlations (through $\mathbf{W}_p$). Note that $\tilde{\mathbf{L}}$, $\mathbf{H}_p$, and $\mathbf{W}_p$ all depend on time through $t_{k_0}$, the most recent encounter before the current time $t$ (as defined in Section 3.2.1). To reflect continuous-time progression, we wrote $\tilde{\mathbf{L}}$ as $\tilde{\mathbf{L}}(t)$: for each ODE integration interval $[t_k, t_{k+1}]$, the Laplacian is constructed from all encounters up to $t_k$ and remains fixed throughout the integration steps within that interval, since no new observations arrive between encounters. Substituting $\tilde{\mathbf{L}}(t)$ into Eq. (4) yields our final knowledge-infused disease progression model:

$$\frac{d\mathbf{S}(t)}{dt} = -\tilde{\mathbf{L}}(t) \left[\mathbf{S}(t) + h(\mathbf{x}(t))\right] \mathbf{\Theta}. \tag{14}$$

The complete training procedure and pseudocode are provided in Appendix B, while the **computational complexity analysis** is detailed in Appendix C.

# 4 EXPERIMENTS

## 4.1 EXPERIMENTAL SETUP

**Datasets.** We conducted experiments on two EHR datasets: (1) a clinical dataset collected from a regional medical network affiliated with our institution, referred to as the *University Hospital* dataset; and (2) *MIMIC-IV* (Johnson et al., 2023), a publicly accessible EHR dataset. We extracted 34 risk factors associated with the progression of diabetes and its complications, as detailed in Table 4 (Appendix D.1), and identified 21 outcome markers of diabetes complications $\mathcal{V}$, as summarized in Table 5 (Appendix D.2). We also constructed a disease progression hypergraph $\mathcal{H}$ based on expert-validated clinical pathways provided by our clinical collaborators, as detailed in Appendix D.3. The *University Hospital* and *MIMIC-IV* datasets contained 2,415 patients and 902 patient sequences, respectively. More details are provided in Appendix D.4.

**Baselines.** To evaluate TD-HNODE, we compared it with representative baselines across four categories. For **Sequential Models**, we included T-LSTM (Baytas et al., 2017), which handles irregular visits using time-aware LSTM, and ContiFormer (Chen et al., 2024b), which redefines self-attention over evolving latent trajectories. For **Temporal Graph Neural Networks**, we used discrete-time MegaCRN (Jiang et al., 2023) and continuous-time TGNE (Cheng et al., 2024), which

Table 1: Results (%, average $\pm$ std) of all methods on *University Hospital* dataset and *MIMIC-IV* dataset, with the best results in bold.

| Methods | University Hospital | | | | MIMIC-IV | | | |
|---|---|---|---|---|---|---|---|---|
| | Accuracy | Precision | Recall | F1-score | Accuracy | Precision | Recall | F1-score |
| T-LSTM | $69.2_{\pm 0.4}$ | $10.6_{\pm 0.2}$ | $55.7_{\pm 0.2}$ | $12.8_{\pm 0.1}$ | $84.1_{\pm 0.2}$ | $17.0_{\pm 0.5}$ | $58.2_{\pm 0.1}$ | $24.5_{\pm 0.3}$ |
| ContiFormer | $77.2_{\pm 0.2}$ | $12.3_{\pm 0.1}$ | $65.4_{\pm 0.2}$ | $16.7_{\pm 0.2}$ | $86.2_{\pm 0.1}$ | $26.2_{\pm 0.4}$ | $82.1_{\pm 0.2}$ | $36.5_{\pm 0.3}$ |
| MegaCRN | $70.7_{\pm 0.1}$ | $8.1_{\pm 0.1}$ | $64.5_{\pm 0.6}$ | $12.9_{\pm 0.1}$ | $83.4_{\pm 0.5}$ | $22.4_{\pm 0.3}$ | $70.1_{\pm 1.3}$ | $30.8_{\pm 0.4}$ |
| TGNE | $72.4_{\pm 0.2}$ | $8.6_{\pm 0.1}$ | $75.4_{\pm 0.3}$ | $14.2_{\pm 0.1}$ | $85.0_{\pm 0.5}$ | $23.4_{\pm 0.1}$ | $72.8_{\pm 0.2}$ | $32.1_{\pm 0.2}$ |
| DHSL | $72.8_{\pm 0.1}$ | $8.3_{\pm 0.1}$ | $60.5_{\pm 0.3}$ | $13.5_{\pm 0.3}$ | $82.9_{\pm 0.2}$ | $22.0_{\pm 0.2}$ | $69.9_{\pm 0.3}$ | $29.8_{\pm 0.2}$ |
| HyperTime | $74.9_{\pm 0.2}$ | $9.3_{\pm 0.1}$ | $59.0_{\pm 0.1}$ | $13.9_{\pm 0.1}$ | $84.6_{\pm 0.2}$ | $25.1_{\pm 0.1}$ | $71.4_{\pm 0.2}$ | $31.7_{\pm 0.2}$ |
| NODE | $72.3_{\pm 0.1}$ | $10.7_{\pm 0.3}$ | $56.9_{\pm 0.1}$ | $14.4_{\pm 0.3}$ | $84.4_{\pm 0.2}$ | $19.5_{\pm 0.3}$ | $62.3_{\pm 0.2}$ | $27.2_{\pm 0.1}$ |
| CODE-RNN | $73.0_{\pm 0.1}$ | $10.0_{\pm 0.1}$ | $61.5_{\pm 0.2}$ | $15.0_{\pm 0.1}$ | $85.7_{\pm 0.1}$ | $23.5_{\pm 0.2}$ | $74.5_{\pm 0.4}$ | $32.5_{\pm 0.2}$ |
| **TD-HNODE** | $\mathbf{79.4}_{\pm \mathbf{0.1}}$ | $\mathbf{14.3}_{\pm \mathbf{0.2}}$ | $\mathbf{79.3}_{\pm \mathbf{0.3}}$ | $\mathbf{20.4}_{\pm \mathbf{0.4}}$ | $\mathbf{87.9}_{\pm \mathbf{0.1}}$ | $\mathbf{31.8}_{\pm \mathbf{0.4}}$ | $\mathbf{85.7}_{\pm \mathbf{0.4}}$ | $\mathbf{42.9}_{\pm \mathbf{1.3}}$ |

Table 2: Ablation study results (%) of TD-HNODE on *University Hospital* and *MIMIC-IV* dataset.

| $\mathbf{H}_p$ | $\mathbf{W}_p$ | University Hospital | | | | MIMIC-IV | | | |
|---|---|---|---|---|---|---|---|---|---|
| | | Accuracy | Precision | Recall | F1-score | Accuracy | Precision | Recall | F1-score |
| ✓ | ✓ | 79.4 | 14.3 | 79.3 | 20.4 | 87.9 | 31.8 | 85.7 | 42.9 |
| ✗ | ✓ | 75.3 | 12.9 | 77.0 | 18.9 | 86.1 | 27.7 | 78.5 | 36.6 |
| ✓ | ✗ | 76.5 | 11.5 | 79.2 | 18.7 | 86.8 | 29.0 | 84.9 | 38.5 |
| ✗ | ✗ | 73.1 | 10.6 | 76.1 | 15.5 | 83.0 | 23.3 | 73.1 | 30.8 |

model temporal graphs via snapshots and event-based messages, respectively. For **Temporal Hypergraph Neural Networks**, we compared with DHSL (Wang et al., 2024a), which fuses hypergraph convolution with GRU, and HyperTime (Younis and Ahmadi, 2024), which builds dynamic hypergraphs with LSTM-enhanced hyperedge convolutions. For **Neural ODE-based Models**, we included NODE (Chen et al., 2018), capturing continuous dynamics in latent space, and CODE-RNN (Coelho et al., 2025), which integrates ODEs with RNNs for time-series modeling. Full baselines implementation details are provided in Appendix D.6.

We evaluated performance using Accuracy, Precision, Recall, and F1-score, emphasizing Recall due to class imbalance and the need to detect early progression. High Recall reflects better identification of true positives, critical for clinical deployment. Source code has been included in the Supplementary Material for reproducibility, and implementation details are provided in Appendix D.5.

## 4.2 COMPARISON ON CLASSIFICATION PERFORMANCE

We compared TD-HNODE with all baselines on the *University Hospital* and *MIMIC-IV* datasets, with results summarized in Table 1. TD-HNODE consistently achieved the best performance across all metrics. On *University Hospital*, it outperformed the strongest baseline (ContiFormer) by 2.2% in accuracy and 3.7% in F1-score, and by 1.7% and 6.4% on *MIMIC-IV*. Compared to non-structural models like T-LSTM, NODE, and CODE-RNN, TD-HNODE showed substantial Recall and F1 gains (*e.g.,* +23.4% Recall over NODE on *MIMIC-IV*), demonstrating its strength in capturing complex disease dynamics through its trajectory-aware hypergraph structure. TD-HNODE also surpassed temporal structure models such as MegaCRN and TGNE. Notably, it achieved 3.9% and 12.9% Recall improvements over TGNE on *University Hospital* and *MIMIC-IV*, respectively. These results confirmed that modeling high-order multi-node interactions via hyperedges offered stronger representational power than traditional pairwise edges, enabling more accurate and clinically meaningful early disease prediction with fewer false negatives.

Additional experiments on cardiovascular disease are provided in Appendix D.7, further demonstrating the generalizability of TD-HNODE beyond diabetes.

## 4.3 ABLATION STUDY

We assessed the impact of two core components in TD-HNODE: the **adaptive incidence matrix** $\mathbf{H}_p$ and the **learnable hyperedge weights** $\mathbf{W}_p$. As shown in Table 2, using $\mathbf{H}_p$ improved F1-score

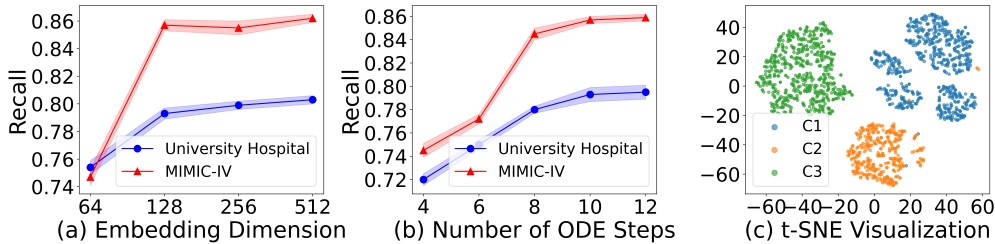

Figure 3: (a) Recall of TD-HNODE on both datasets with varying embedding dimensions. (b) Recall with varying numbers of ODE steps. (c) t-SNE visualization of 1,690 patients from the *University Hospital* dataset; C1, C2, and C3 denote Clusters 1, 2, and 3.

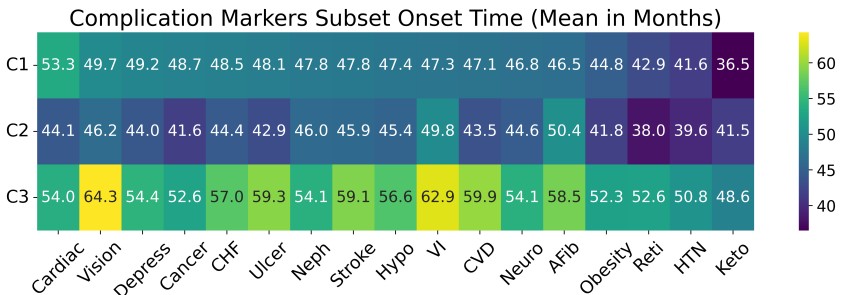

Figure 4: Mean onset time (in months from the first encounter) of each marker across patients in each cluster. Earlier values indicate earlier manifestation or faster progression.

from 15.5% to 18.7% without $\mathbf{W}_p$, and to 20.4% with it, highlighting the benefit of intra-trajectory attention for time-aware marker importance. Meanwhile, enabling $\mathbf{W}_p$ boosted recall from 76.1% to 77.0% and F1-score from 15.5% to 18.9% even with static $\mathbf{H}$, and yielded 5.4%/5.8% gains on recall/F1 on *MIMIC-IV*, validating the value of modeling inter-trajectory dependencies.

## 4.4 SENSITIVITY ANALYSIS

We analyzed TD-HNODE's sensitivity to two hyperparameters: embedding dimension $d$ and the number of ODE solver steps in RK4. As shown in Figure 3(a), increasing $d$ from 64 to 128 significantly improved recall (*e.g.,* from 0.747 to 0.857 on *MIMIC-IV*), but further increases showed diminishing returns or overfitting, so we chose $d = 128$. Similarly, Figure 3(b) shows that varying the number of steps from 4 to 12 revealed underfitting at lower values (*e.g.,* 4 or 6), while performance stabilized around 10 steps with minimal gains beyond. These results confirmed TD-HNODE's robustness and the suitability of its default hyperparameter settings for real-world EHR applications.

## 4.5 CASE STUDY

We conducted a case study to apply the TD-HNODE results for patient progression sub-phenotyping. Specifically, we extracted the patient embeddings (prior to the decoder layers) learned from TD-HNODE on the *University Hospital* dataset. We visualized the patient embeddings in 2D using t-SNE projection and identified three clear clusters, as shown in Figure 3(c). We then used hierarchical clustering to group patients into these three clusters and analyzed disease progression patterns within each cluster. Specifically, for each of the 21 complication markers, we computed the *mean onset time* (in months since the first encounter) for patients within each cluster. A smaller onset time indicated faster progression of complication outcomes. As shown in Figure 4, patients in Cluster 3 exhibited the slowest progression, followed by Cluster 1, and then Cluster 2, which showed the most rapid progression. For example, compared to Cluster 3, patients in Cluster 2 experienced earlier onset by 9 months in *Cardiac Revascularization (Cardiac)*, 18 months in *Blindness and Vision Loss (Vision)*, and 12 months in *Congestive Heart Failure (CHF)*. These results demonstrated that TD-HNODE effectively captures the heterogeneity within the patient cohort.

## 5 CONCLUSION AND FUTURE WORK

In this work, we proposed TD-HNODE, a novel framework for continuous-time disease progression modeling that integrates medical knowledge with a TD-hypergraph-based Neural ODE. The method captures both intra- and inter-trajectory dependencies, and experiments on real-world EHR datasets demonstrated its effectiveness in modeling diabetes progression. For future work, our framework currently focuses on known disease progression pathways, as in chronic diseases such as diabetes, and should be extended to infer unknown or partially characterized trajectories (*e.g.,* via frequent pattern mining or Bayesian networks). In addition, we plan to incorporate causal inference to evaluate the impact of complex treatment regimens.

## ACKNOWLEDGMENTS

This material is based upon work supported by the National Science Foundation (NSF) under Grant No. IIS-2147908, IIS-2207072, OAC-2152085, OAC-2402946, and OAC-2410884.

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

APPENDIX

## A   MATHEMATICAL NOTATIONS

The main mathematical notations of this paper are shown in Table 3.

Table 3: Main mathematical notations.

| Notation | Description |
|---|---|
| $\mathbf{x}_u(t_k)$ | Risk factors vector of patient $u$ at time $t_k$ |
| $\mathbf{y}_u(t_k)$ | Marker status vector of patient $u$ at time $t_k$ |
| $\mathcal{V}$ | Set of markers (nodes) |
| $p_j$ | The $j$-th trajectory: $< v_1^j, v_2^j, ..., v_{|p_j|}^j >$ |
| $n$ | Number of predefined markers (nodes) |
| $m$ | Number of predefined trajectories (hyperedges) |
| $v_i^j$ | The $i$-th marker in trajectory $p_j$ |
| $\mathcal{H}$ | Disease progression hypergraph: $(\mathcal{V}, \mathcal{E})$ |
| $p_j^u$ | Patient $u$'s temporally detailed trajectory along $p_j$ |
| $\mathcal{H}^u$ | TD-Hypergraph of patient $u$: $(\mathcal{V}, \mathcal{E}^u)$ |
| $e_j^u$ | TD-Hyperedge of $u$ along $p_j$ |

## B    TRAINING ALGORITHM

After integrating the ODE from the latest observed timestamp $t_k$ to $t_{k+1}$, we obtain $\mathbf{S}(t_{k+1}) \in \mathbb{R}^{n \times d}$, the hidden representations of all $n$ markers at time $t_{k+1}$. These embeddings are then mapped to prediction scores for each marker, followed by a sigmoid activation to estimate the probability of presence for each marker at $t_{k+1}$. The model is trained by minimizing the binary cross-entropy (BCE) loss between the predicted probabilities and the ground-truth marker statuses.

The overall training process is detailed in **Algorithm 1** below. In particular, we adopt an auto-regressive training loop: for each patient, the model integrates the continuous dynamics forward in time, using the hidden state at $t_k$ to predict marker outcomes at $t_{k+1}$. At each step, the TD-Hypergraph Laplacian computed at time $t_k$ guides the flow of information, capturing fine-grained trajectory dynamics and marker dependencies.

---

**Algorithm 1** Training procedure of TD-HNODE

---

**Require:** • Encounter sequences for $N$ patients:

$$\{\mathbf{x}_u(t_1), \ldots, \mathbf{x}_u(t_{l_u}); \ \mathbf{y}_u(t_1), \ldots, \mathbf{y}_u(t_{l_u})\}_{u=1}^N ;$$

   • TD-Hypergraphs: $\{\mathcal{H}^u\}_{u=1}^N$
**Ensure:** Model parameters $\boldsymbol{\Theta}$
1:   Initialize all parameters $\boldsymbol{\Theta}$
2:   **for** epoch in 1 : MaxEpoch **do**
3:     **for** each patient $u$ **do**
4:       Initialize hidden state $\mathbf{S}_u(t_1)$
5:       **for** timestamp $t_k \leftarrow t_1 : t_{l_u}$ **do**
6:         Compute incidence matrix $\mathbf{H}_p(t_k)$ (Eq. 10)
7:         Compute hyperedge weights $\mathbf{W}_p(t_k)$ (Eq. 12)
8:         Compute TD-Hypergraph Laplacian $\tilde{\mathbf{L}}(t_k)$ (Eq. 13)
9:         Update state: $\mathbf{S}_u(t_{k+1}) \leftarrow \text{ODESolver}(\mathbf{S}_u(t_k), \mathbf{x}_u(t_k), \tilde{\mathbf{L}}(t_k), [t_k, t_{k+1}])$
10:        $\hat{\mathbf{y}}_u(t_{k+1}) = \text{Sigmoid}(\text{Dense}(\mathbf{S}_u(t_{k+1})))$
11:        Compute loss $\mathcal{L} \leftarrow \text{BCE}(\hat{\mathbf{y}}_u(t_{k+1}), \mathbf{y}_u(t_{k+1}))$
12:      **end for**
13:    **end for**
14:    Update $\boldsymbol{\Theta}$ via gradient descent on accumulated loss
15:  **end for**
16:  **return** $\boldsymbol{\Theta}$

---

## C  COMPUTATIONAL COMPLEXITY

Assume the average number of timestamps per patient sequence is $\hat{L}$, the batch size is $B$, the number of markers (nodes) is $n$, the number of hyperedges is $m$, the number of attention heads is $\hat{h}$, and the hidden dimension is $d$. We denote the per-head dimension as $d_h = d/\hat{h}$, the average number of markers per hyperedge as $n_e$, and the number of Runge-Kutta steps in Neural ODE as $s$.

- Attention-based incidence matrix (Eq. 9): With multi-head attention, each head operates on dimension $d_h$. The per-head query dimensions are $Q \in \mathbb{R}^{B \times m \times d_h}$ and the key dimensions are $K \in \mathbb{R}^{B \times m \times n_e \times d_h}$. The dot-product attention computation across all $\hat{h}$ heads has complexity $O(B \cdot m \cdot n_e \cdot d)$ per timestamp.

- Hyperedge weight matrix (Eq. 12): Self-attention pooling within each hyperedge costs $O(B \cdot m \cdot n_e^2 \cdot d)$. The linear projection $\tilde{\mathbf{G}} = \mathbf{G}\,\mathbf{W}_{\mathcal{E}}$ costs $O(B \cdot m \cdot d^2)$, and computing the trajectory correlation matrix $\mathbf{W}_p = \tilde{\mathbf{G}}\,\tilde{\mathbf{G}}^\top \in \mathbb{R}^{m \times m}$ costs $O(B \cdot m^2 \cdot d)$. Total complexity is $O(B \cdot (m \cdot n_e^2 \cdot d + m \cdot d^2 + m^2 \cdot d))$.

- TD-Hypergraph Laplacian (Eq. 13): The Laplacian $\tilde{\mathbf{L}} \in \mathbb{R}^{n \times n}$ is constructed once per timestamp and reused across ODE steps. The dominant costs are the matrix products $\mathbf{H}_p\mathbf{W}_p$ at $O(n \cdot m^2)$ and the subsequent product with $\mathbf{H}_p^\top$ at $O(n^2 \cdot m)$. Total construction cost is $O(B \cdot (n \cdot m^2 + n^2 \cdot m))$ per timestamp.

- ODE solver: Each Runge-Kutta step applies the precomputed $\tilde{\mathbf{L}} \in \mathbb{R}^{n \times n}$ to $[\mathbf{S}(t) + h(\mathbf{x}(t))] \in \mathbb{R}^{n \times d}$, costing $O(n^2 \cdot d)$, and multiplies by the projection matrix $\mathbf{\Theta} \in \mathbb{R}^{d \times d}$, costing $O(n \cdot d^2)$. Over $s$ steps, the total cost is $O(B \cdot s \cdot n \cdot d \cdot (n + d))$ per timestamp.

The overall per-batch complexity of TD-HNODE is

$$T_{\text{total}} = O\Big(B \cdot \hat{L} \cdot \big(mn_e d + mn_e^2 d + md^2 + m^2 d + nm^2 + n^2 m + s\,n\,d\,(n+d)\big)\Big),$$

and since $n_e \le n$, we can further bound it by

$$T_{\text{total}} \le O\big(B \cdot \hat{L} \cdot (mn^2 d + md^2 + s\,n\,d^2)\big).$$

Given our settings with $n \in [20, 30]$, $m \in [10, 15]$, $d = 128$, and $\hat{L} \le 20$, the dominant cost is the ODE solver $O(B \cdot \hat{L} \cdot s \cdot n \cdot d^2)$, and overall complexity scales linearly with the sequence length $\hat{L}$, the number of markers $n$, and the number of ODE steps $s$.

# D EXPERIMENTS

## D.1 RISK FACTORS

The risk factors were defined below by our clinical co-author (MD) based on established expertise and supported by the literature on diabetes complications (Tomic et al., 2022).

Table 4: List of 34 risk factors used in the study.

| | |
|---|---|
| SEX_CD | GFR |
| HDL | Triglycerides |
| Beta_blockers | CCB |
| DPP4i | Lipid |
| Loop | Metformin |
| Non_loop | RAS |
| Sulfonylurea | Thiazolidinedione |
| Alcohol_use_disorder | Angina_flag |
| Cardiovascular_Disease | Chronic_kidney_disease |
| Drug_use_disorder | End_Stage_Renal_Disease |
| Exercise | Gestational_diabetes |
| History_of_Myocardial_Infarction | History_of_stroke |
| HIV_AIDS | Hypercholesterolaemia |
| Lower_extremity_amputation | Myocardial_Infarction_MI |
| Organ_transplant | Peripheral_vascular_disease |
| Photocoagulation | Pregnancy |
| Retinopathy_intravitreal_injections | Secondary_diabetes |

## D.2 COMPLICATION MARKERS

We listed a detailed table reporting the positive prevalence (%) for each of the 21 complication markers on both datasets (shown below Table 5). The 21 diabetes complication outcome markers were defined by our clinical co-author (MD) based on established clinical expertise and supported by the literature on diabetes complications (Tomic et al., 2022). In our formulation, we treat all 21 markers as **irreversible**. Although some markers such as 'HbA1c Low', 'HbA1c High', 'Poor Lipid', and 'Poor BP' are laboratory test results that may fluctuate over time, we emphasize their *first occurrence* as an indication that the patient has progressed to this stage of disease or has experienced this level of abnormality. Following clinical guidance, we therefore model the initial onset of each marker as a significant progression event that remains active in the disease trajectory representation.

Table 5: Complication marker prevalence in *University Hospital* and *MIMIC-IV*.

| Complication Markers | University Hospital | MIMIC-IV |
|---|---|---|
| HbA1c Low | 0.058 | 0.028 |
| Poor Lipid | 0.026 | 0.039 |
| Hypertension (HTN) | 0.068 | 0.043 |
| Obesity | 0.058 | 0.022 |
| Foot Ulcer (Ulcer) | 0.005 | 0.001 |
| Blindness and Vision Loss (Vision) | 0.003 | 0.001 |
| Visual Impairment (VI) | 0.001 | 0.001 |
| Heart Failure (CHF) | 0.018 | 0.024 |
| Nephropathy (Neph) | 0.030 | 0.026 |
| Neuropathy (Neuro) | 0.051 | 0.018 |
| Retinopathy (Reti) | 0.011 | 0.006 |
| Cerebrovascular Disease (CVD) | 0.011 | 0.002 |
| Stroke | 0.011 | 0.003 |
| Depression (Depress) | 0.026 | 0.038 |
| Hypoglycemia (Hypo) | 0.022 | 0.003 |
| HbA1c High | 0.053 | 0.018 |
| Poor BP | 0.024 | 0.005 |
| Cardiac Revascularization (Cardiac) | 0.009 | 0.010 |
| Atrial Fibrillation (AFib) | 0.007 | 0.021 |
| Cancer | 0.014 | 0.001 |
| DKA (Keto) | 0.002 | 0.019 |
| Average | **0.024** | **0.016** |

### D.3 DISEASE PROGRESSION PATHWAYS

The predefined hyperedges (progression pathways) were determined below with guidance from our clinical co-author (MD) and supported by established medical literature (Fonseca, 2009; Yu et al., 2024).

- HbA1c High → Poor Lipid → Hypertension / Poor BP → Atrial Fibrillation → Heart Failure
- HbA1c High → Obesity
- HbA1c High → Retinopathy → Visual Impairment → Blindness and Vision Loss
- HbA1c Low → Hypoglycemia
- HbA1c High → DKA
- HbA1c High → Poor Lipid → Hypertension / Poor BP → Cardiac Revascularization
- HbA1c High → Depression
- HbA1c High → Poor Lipid → Hypertension / Poor BP → Cerebrovascular Disease → Stroke
- HbA1c High → Neuropathy → Foot Ulcer
- HbA1c High → Nephropathy

### D.4 DATASET DETAILS

We provide additional details about the two datasets used in our study.

**University Hospital Dataset.** This study was approved by the Institutional Review Board (IRB) of our institution. This dataset contains longitudinal EHR sequences for 2,415 diabetic patients collected from a regional medical network affiliated with our institution. Each patient record consists of a time-ordered sequence of hospital encounters, including structured clinical information such as laboratory test results, vital signs, medications, and diagnosis codes.

**MIMIC-IV Dataset.** MIMIC-IV (Johnson et al., 2023) is a publicly available EHR dataset collected from the Beth Israel Deaconess Medical Center. Although not specifically designed for diabetes, we identified 902 patients with at least one diabetes-related complication by mapping ICD-9/10 diagnosis codes to our predefined marker set $\mathcal{V}$.

**Preprocessing.** For both datasets, we organized each patient's data as a sequence of 20 encounters, with each encounter containing a risk factor vector $\mathbf{x}_u(t_k)$ and a binary complication marker vector $\mathbf{y}_u(t_k)$. For the University Hospital dataset, the raw encounter sequence lengths ranged from 10 to 40. To standardize the input format, we fixed the sequence length at 20. The basic statistics of both datasets are shown in Table 6. For patients with fewer than 20 encounters, we applied padding at the end of the sequence; for those with more than 20, we selected the latest 20 encounters, as disease progression events tend to occur in later stages of follow-up.

Table 6: Dataset statistics for University Hospital and MIMIC-IV.

| Dataset | Number of Encounters | | | Time Span (Months) | | |
|---|---|---|---|---|---|---|
| | Min | Avg | Max | Min | Avg | Max |
| University Hospital | 10 | 15 | 40 | 19.4 | 68.6 | 121.7 |
| MIMIC-IV | 20 | 20 | 20 | 6.4 | 73.1 | 177.1 |

Missing values in the risk factor vectors were imputed using the most recent non-missing value from prior encounters (i.e., last-observation carried forward). This approach preserves temporal consistency and aligns with clinical practice, where outdated test results are often referenced until updated.

In addition, under guidance from clinical collaborators, several continuous-valued physiological indicators were discretized into clinically meaningful categories. Specifically:

- **GFR** values were discretized into `GFR_NORM` ($\geq 90$), `GFR_Decrease_Slight` ($60 \leq$ GFR $< 90$), and `GFR_Decrease_Severe` ($< 60$).

- **HDL** values were mapped to `HDL_Good` ($\geq 60$), `HDL_Normal` ($40 \leq$ HDL $< 60$), and `HDL_Bad` ($< 40$).

- **Triglycerides** were categorized into `Triglycerides_Good` ($< 150$), `Triglycerides_LowRisk` ($150 \leq$ TG $< 199$), and `Triglycerides_HighRisk` ($\geq 199$).

These discrete categories were embedded as input tokens for our model.

For the MIMIC-IV dataset, only structured diagnosis codes were used to construct the complication marker vectors $\mathbf{y}_u(t_k)$ by mapping ICD-9/10 codes to the predefined marker set $\mathcal{V}$. Risk factor inputs were not utilized due to sparsity and inconsistency across patient records.

**Hypergraph Construction.** We constructed a disease progression hypergraph $\mathcal{H}$ based on expert-validated trajectories of diabetes complications. Each trajectory forms a hyperedge capturing high-order progression patterns among markers, serving as the structural backbone for constructing patient-specific TD-Hypergraphs in our framework.

### D.5 Implementation Details

We applied the same label preprocessing strategy to both datasets: **for each complication marker, only its first occurrence (onset) was used**. This transformed the problem from simple state classification to the actual challenge, i.e., **disease onset prediction** — whether a new complication would occur in the next encounter. For both datasets, we randomly split patients into training, validation, and test sets using an 8:1:1 ratio.

We implemented all models using PyTorch and conducted training on a cluster of $8 \times$NVIDIA A100 GPUs (80GB). Our TD-HNODE model was trained using the Adam optimizer with a learning rate of 1e-4, weight decay of 1e-6, and a batch size of 1. Training was run for up to 200 epochs, with early stopping applied based on the validation loss (patience = 5).

We used 128-dimensional embeddings for both discrete token inputs and continuous time position embedding. The hyperedge attention encoder consisted of 2 attention layers with 8 attention heads each, a feed-forward expansion factor of 4, GELU activation, and dropout rate of 0.1. Positional encodings were learnable, and token embeddings were initialized using a uniform distribution.

The Neural ODE module modeled continuous-time latent dynamics over the TD-Hypergraph using a fixed-step Runge-Kutta 4th-order method (RK4) with 10 solver steps. The input to the ODE consisted of the hypergraph-enhanced representations obtained from the TD-Hypergraph Laplacian, and its outputs were decoded into complication marker probabilities via a sigmoid layer.

Source code has been included in the Supplementary Material for reproducibility.

### D.6 BASELINES DETAILS

To evaluate the effectiveness of TD-HNODE, we compare it against representative baselines from several major categories:

**(1) Sequential Models:**

- **T-LSTM** Baytas et al. (2017): A time-aware LSTM variant designed to handle irregularly sampled patient records.
- **ContiFormer** Chen et al. (2024b): Extends the Transformer architecture by redefining self-attention to operate over time-evolving latent trajectories.

**(2) Temporal Graph Neural Networks:**

- **MegaCRN** Jiang et al. (2023): A discrete-time GNN model that combines graph convolution with gated recurrent units (GRU), operating on temporal graphs represented as discrete-time snapshots.
- **TGNE** Cheng et al. (2024): A continuous-time GNN that constructs event-based structural messages to model evolving temporal graphs in a fine-grained manner.

**(3) Temporal Hypergraph Neural Networks:**

- **DHSL** Wang et al. (2024a): Combines a hypergraph convolution network with a GRU to jointly model high-order spatial correlations and temporal dependencies.
- **HyperTime** Younis and Ahmadi (2024): Constructs dynamic hypergraphs from time series segments and applies LSTM-enhanced hyperedge convolutions to model evolving temporal patterns.

**(4) Neural ODE-based Models:**

- **NODE** Chen et al. (2018): A foundational continuous-time model that captures smooth temporal dynamics via ODE-based latent evolution.
- **CODE-RNN** Coelho et al. (2025): Combines Neural ODEs with recurrent networks to model temporal dynamics in time-series data.

As for DHSL and HyperTime, both adopt a discrete-time hypergraph neural network: They divide continuous time domain into discrete time intervals (different snapshots). Within each time interval, they construct a static hypergraph snapshot and use hypergraph convolution to extract spatial structure. Then, a recurrent module (GRU/LSTM) is applied across the snapshots to propagate features over time. This design may encounter difficulties in modeling the subtle temporal progression patterns on irregular time data (as in our case).

To ensure fairness, we customized each method to our task setting. Specifically, for models without structural inputs, *i.e.,* Categories (1) and (4), we concatenate the risk factor vector $\mathbf{x}$ and marker status vector $\mathbf{y}$ as inputs. For graph-based models, *i.e.,* Categories (2), we break the TD-Hypergraph $\mathcal{H}$ into a standard graph with pairwise edges. For models that assume discrete time, *i.e.,* MegaCRN, DHSL, HyperTime, we segment the temporal graph or hypergraph into fixed-length snapshots by treating each patient encounter as a separate timestamp. That is, we construct a static graph (or hypergraph) for each encounter and stack them sequentially as discrete time steps. For graph-based models that assume continuous time, *i.e.,* TGNE, we treat each encounter as an individual event, ordered by its timestamp. We convert the temporally detailed hypergraph into a sequence of timestamped edges.

### D.7 ADDITIONAL EXPERIMENTS ON CHRONIC DISEASE

To further evaluate the generalizability of our framework, we incorporated an **additional chronic disease (cardiovascular disease)** using the publicly available MIMIC-IV dataset Johnson et al. (2023). ICD-9/10 codes were mapped to five clinically recognized markers: Hypertension, Atrial Fibrillation, Heart Failure, Cerebrovascular Disease / Stroke, and Myocardial Infarction. Based on prior clinical studies Dzeshka et al. (2015); Bonow et al. (2011), we defined three representative progression pathways:

- Hypertension → Atrial Fibrillation → Heart Failure
- Hypertension → Myocardial Infarction → Heart Failure
- Hypertension → Cerebrovascular Disease / Stroke

This preprocessing resulted in 1,665 patients (train/validation/test = 1,332/166/167), with dataset statistics summarized in Table 7.

Table 7: Statistics of the cardiovascular disease dataset.

| Dataset | Number of Encounters | | | Time Span (Months) | | |
|---|---|---|---|---|---|---|
| | Min | Avg | Max | Min | Avg | Max |
| Cardiovascular Disease | 9 | 14 | 20 | 4.2 | 53.3 | 94.2 |

We compared TD-HNODE with all baseline methods, and the results are presented in Table 8.

Table 8: Performance comparison on the cardiovascular disease dataset.

| Model | Accuracy | Precision | Recall | F1-score |
|---|---|---|---|---|
| T-LSTM | 0.701 | 0.126 | 0.613 | 0.179 |
| ContiFormer | 0.796 | 0.189 | 0.765 | 0.266 |
| MegaCRN | 0.744 | 0.167 | 0.759 | 0.233 |
| TGNE | 0.761 | 0.175 | 0.801 | 0.247 |
| DHSL | 0.740 | 0.150 | 0.717 | 0.232 |
| HyperTime | 0.752 | 0.160 | 0.734 | 0.231 |
| NODE | 0.786 | 0.152 | 0.689 | 0.227 |
| CODE-RNN | 0.779 | 0.166 | 0.699 | 0.190 |
| TD-HNODE | **0.804** | **0.193** | **0.818** | **0.291** |

As shown in Table 8, TD-HNODE consistently outperforms all baselines on the cardiovascular disease dataset, particularly in recall and F1-score. These findings demonstrate that our framework effectively captures the progression of chronic diseases beyond diabetes and highlight its potential for generalization when expert-defined progression pathways are available.

