# OpenReview forum: "Temporally Detailed Hypergraph Neural ODE for Disease Progression Modeling"
_ICLR.cc/2026/Conference — ICLR 2026 Poster_

### Official Review · Reviewer_F3q3 · 2025-10-29

**Soundness:** 3
**Presentation:** 2
**Contribution:** 3
**Rating:** 6
**Confidence:** 5

**Summary:**

This study introduces a novel method that combines Neural ODE and GNN–based for interpretable trajectory generation.
This work uses hypergraph neural network to capture relational dependencies among variables and uses neural ODE to learn continuous dynamics.
The authors further provide theoretical support ensuring representational validity.
Experiments demonstrate that the method achieves accurate trajectory reconstruction and offers interpretable latent dynamics.

**Strengths:**

1. It combines Neural ODEs and GNNs to effectively capture both temporal and spatial dependencies.

2. It improves over the traditional fixed hyperedge weight matrix by using a learnable hyperedge weight matrix.

3. By modeling continuous dynamics and graph relations, the model offers interpretable latent representations and visualizable trajectories that help in analyzing its validity.

4. The experimental section shows superior reconstruction and forecasting accuracy compared with baseline methods.

5. It provides visualization for the generated trajectories.

**Weaknesses:**

1. This paper lacks a comprehensive investigation of Transformer based model for irregularly-sampled event sequence.

For Transformer models, it should include but not limited:

[1] Learning the natural history of human disease with generative transformers. Nature 2025.

For linear Transformer, it can be considered as discreted ODE, for instance:

[2] TrajGPT: Irregular Time-Series Representation Learning of Health Trajectory. IEEE J-BHI 2025.

I just list few recent works. You could find more related works which should be included in the literature review.

2. Although this paper claims that it can generate interpretable trajectory, it has limited data analysis about the generated trajectories. It should include more case studies about how a patient's health progress over time. It could also include population-level analysis about the subphenotypes or combritidy.

3. It lacks interpretation and visualization of the learned token embedding (like HP, AF). We do not know whether model learns meaingful embedding or not in the graph-based model.

**Questions:**

Extending the weakness 2.

1.  While the motivations mentions subpheontypes, did the author try to analyze it? whether it is connected the different progression speed in Fig.4.

2. This work focuses on T2D and selects features to analyze it. Did the author analyze the correlation or combritidy between T2D and other features? whether T2D will also contribute to other diseases (like heart failure)

3. While the background mentions medication and talks about "timely treatment", it does not have any thing about the interactions between disease progression and medications? we should see medication helps the disease recovery?

4. Some symptoms are irreversible. I am wondering whether the medictions can help stabalize it? It is interesting that the authors mention irreversible symptoms but it does not make analysis in the result section.

---

> ### Author Response · Authors · 2025-11-25
> **Response to Reviewer F3q3**
>
> We sincerely thank the reviewer F3q3 for the positive feedback and we address your comments below. **(W = Weakness, Q = Question)**.
>
> **W1. Transformer-based Baselines.**
>
> Thank you for the suggestion. Paper [1] and [2] were published around or after the ICLR submission deadline. We will add them in the revised Related Work section. Specifically, these two works are Transformer-based models that approach disease progression primarily as **sequence modeling**: [1] leverages population-scale generative pre-training, while [2] utilizes ODE-based attention to handle irregular sampling. They do not explicitly model the hypergraph structure of progression markers.
>
> [1] Learning the natural history of human disease with generative transformers. Nature 2025.
> [2] TrajGPT: Irregular Time-Series Representation Learning of Health Trajectory. IEEE J-BHI 2025.
>
>
> **W2 & Q1 & Q2. Subphenotypes, and Progression Speed.**
>
> Thanks for the suggestion. In fact, we already have a case study of patient subphenotyping (see Section 4.5 and Figures 3(c) & 4). Specifically, we performed hierarchical clustering on the patient embeddings learned from TD-HNODE and identified three clear patient clusters (subphenotypes) in Figure 3(c). We also compared these clusters by visualizing their disease progression pace. As shown in Fig. 4, these clusters clearly show different progression speeds: Cluster 2 corresponds to rapid progressors with the earliest complication onsets (e.g., Vision Loss ~18 months earlier), while Cluster 3 represents slow progressors with consistently delayed onsets across markers.
>
> We agree with the reviewer that population-level comorbidity/correlation analysis between T2D and other conditions (e.g., heart diseases). We will explore this in future work.
>
>
>
> **W3. Visualization of Token Embeddings.**
>
> Thank you for this suggestion. We will add a compact embedding-level visualization (e.g., a marker–marker similarity heatmap and a 2D projection such as UMAP/t-SNE) to demonstrate that learned token embeddings reflect similarity between markers in the clinical domain.
>
> **Q3 & Q4. Medications and Stability.**
>
> This is a great comment! We fully agree with the reviewer that studying interactions between disease progression and medications is very important. In fact, this is why we started this work. We are currently working on using causal machine learning to study the different treatment effects of medications on learned patient sub-phenotypes. This will be presented in a follow-up paper.

---

### Official Review · Reviewer_wNfJ · 2025-10-29

**Soundness:** 2
**Presentation:** 1
**Contribution:** 2
**Rating:** 2
**Confidence:** 4

**Summary:**

This paper introduces TD-HNODE, a disease progression model that integrates clinical knowledge into a temporally detailed hypergraph combined with Neural ODE. Each node represents a disease complication marker, and each hyperedge is a predefined clinically validated progression trajectory. For disease modeling, the authors propose a time-adaptive Laplacian that governs continuous-time diffusion of latent marker states, comprising an attention-based incidence matrix for patient-specific, time-aware weighting of markers and a learnable hyperedge weight matrix. Experiments on two EHR datasets (University Hospital and MIMIC-IV) show that TD-HNODE improves accuracy, recall, and F1 compared with strong baselines (T-LSTM, ContiFormer, TGNE, HyperTime, CODE-RNN). Ablations support the contributions of both adaptive incidence $H_p$ and learnable weights $W_p$, and a case study suggests that the model’s latent embeddings reveal sub-phenotypes of diabetic progression.

**Strengths:**

1. The topic is important. Continuous-time modeling of chronic disease trajectories is an important and emerging topic.
2. Interpretability might be good, as the hyperedges align with known clinical pathways. This might serve as a foundation for explanation and clinical validation.
3. Consistent improvements on two EHR datasets, particularly in recall (clinically critical for early detection).

**Weaknesses:**

1. Over-complex and arguably unnatural construction.
While the idea of embedding medical knowledge into continuous-time dynamics is valuable, the resulting architecture feels heavily engineered. TD-HNODE stacks many modeling layers—curated trajectories $\rightarrow$ hypergraph $\rightarrow$  attention-based incidence $\rightarrow$  dense inter-trajectory weighting $\rightarrow$  ODE integration—each adding parameters without clear generative justification. It is difficult to discern whether the model captures meaningful structure or merely benefits from large capacity.
2. Everything is learnable, risking loss of inductive bias. Almost all structural components ($H_p$, $W_p$, time encodings) are trainable.
This undermines the “knowledge-infused” motivation: the learned Laplacian may diverge from curated pathways, reducing interpretability.
Regularization toward clinical priors or partial parameter freezing would help maintain domain grounding.
3. Bias compounding across multiple submodules.
The architecture effectively stacks a model on top of another (attention $\rightarrow$ self-attention $\rightarrow$ pooling $\rightarrow$ ODE $\rightarrow$ decoder). Each stage may introduce its own bias, and the composition could amplify rather than mitigate error.
It remains unclear which layer drives performance versus redundancy.
4. Questionable scalability of the hypergraph construction.
The temporally detailed hypergraph may require combining all observed markers across trajectories, potentially leading to a combinatorial explosion of hyperedges. The paper does not quantify computational or memory costs beyond brief remarks in the appendix.
Recomputing dense, time-varying $\tilde{L}(t)$ could be infeasible for large EHRs.
5. Clarity and naturalness of formulation. The path from clinical intuition to mathematical formulation is hard to follow. Although mathematically consistent, the paper could better motivate why this sequence of modeling choices is natural or necessary.

**Questions:**

1. Please detail how predefined trajectories were constructed (clinical source, inter-rater agreement, # of trajectories, examples).
2. Since hyperedges are set-valued, how is order encoded beyond attention?
3. Why not construct trajectories from actual temporally detailed sequences (e.g., frequent pattern mining, partial order mining), then regularize by clinical priors?
4. Classical hypergraph Laplacians rely on diagonal W to ensure symmetry and PSD. With dense $W_p$, $\tilde L$ may not be symmetric/PSD unless extra constraints hold. Non-PSD can destabilize diffusion and the ODE. Please prove conditions under which $\tilde L(t)$ is symmetric/PSD with dense $W_p$.
5. Please specify the exact initialization: $S(t_1)=\mathrm{Enc}([x(t_1), y(t_1)])$

---

> ### Author Response · Authors · 2025-11-30
> **Response to Reviewer wNfJ**
>
> **W1 & W3. Complexity and Redundancy**
>
> **We want to clarify that the comment** that “TD-HNODE stacks many modeling layers—curated trajectories $\to$ hypergraph $\to$  attention-based incidence $\to$  dense inter-trajectory weighting $\to$ ODE integration—each adding parameters without clear generative justification” **is inaccurate**.
>
> In fact, **curated trajectories $\to$ hypergraph are not our modeling layers; they are part of data preprocessing**. The layers of the model with trainable parameters are only the last three, i.e., attention-based incidence, inter-trajectory weighting, and ODE integration. As the reviewer asked, to discern whether each module captures meaningful structure or merely benefits from large capacity, we already did an **ablation study**. Our ablation study (Table 2) explicitly shows that these model components are not redundant: for example, removing the adaptive incidence $\mathbf{H}_p$ or the learnable weights $\mathbf{W}_p$ significantly degrades performance (e.g., F1 drops from 42.9% to 30.8% on MIMIC-IV).
>
>
> **W2. Learnability vs. Inductive Bias**
>
> We respectfully clarify that **not everything is learnable**. In fact, our design of the learnable TD-Hypergraph Laplacian contains hard-coded trajectory knowledge (e.g., the binary incidence matrix $\mathbf{H}$). For instance, the adaptive incidence is defined as $\mathbf{H}_p(i,e) = \mathbf{H}(i,e) \cdot \alpha_e$ in Eq. 7, where $\mathbf{H}$ is non-trainable, and each non-zero entry $\mathbf{H}(i,e)=1$ comes directly from the curated hypergraph trajectories.
> Thus, mathematically, $\mathbf{H}_p$ strictly preserves the non-zero pattern of $\mathbf{H}$: if $\mathbf{H}(i,e)=0$, then $\mathbf{H}_p(i,e)=0$. The only learnable part is the magnitude of the non-zero entries through $\alpha_e$. In other words, $\mathbf{H}_p$ can only reweight existing clinical connections but can never create new edges or change the hypergraph structure. Therefore, our design already ensures that **clinical priors are explicitly encoded**.
>
>
> **W4. Scalability (Hyperedge Explosion)**
>
> **This appears to be a misunderstanding**. Our hyperedges are predefined only on the set of clinically verified disease progression trajectories, instead of all possible combinations. They **do not include combinations of markers that do not make clinical sense**. Therefore, the total number is far below exponential.
>
>
> **Q1 & Q3. Trajectory Construction**
>
> **We explicitly listed every trajectory in Appendix D.3 of the original submission**. In fact, our trajectory construction process follows the same process as mentioned by the reviewer in Q3. We employed a rigorous two-step approach: first, we identified candidate progression patterns from the EHR data using frequent pattern mining; second, these candidates were validated by our clinical co-author (with an MD degree) in reference to medical literature (e.g., [1, 2]). This process yielded 13 clinically verified trajectories.
>
> [1] Yu, Marc Gregory, Daniel Gordin, Jialin Fu, Kyoungmin Park, Qian Li, and George Liang King. "Protective factors and the pathogenesis of complications in diabetes." Endocrine reviews 45, no. 2 (2024): 227-252.
>
> [2] Fonseca, Vivian A. "Defining and characterizing the progression of type 2 diabetes." Diabetes care 32, no. Suppl 2 (2009): S151.
>
> **Q2. Order Encoding**
>
> As defined in Section 2.1, our "Temporally Detailed Hyperedge" is **not a set, but an ordered sequence** of tuples $(v_i^j, t_i)$. The timestamps $t_i$ and the sequence order are explicitly encoded via continuous-time positional embeddings in the attention mechanism, as shown in Eq. 4 and 5.
>
> **Q4. Symmetric/PSD Condition**
>
> In our implementation, the fixed constant degree matrix $\mathbf{D}_e^{-1}$ is absorbed into the parameterized hyperedge weight matrix $\mathbf{W}_p$. Specifically, we parameterize this weight matrix directly as $\mathbf{W}_p = \tilde{\mathbf{G}}\tilde{\mathbf{G}}^T$ (as defined in Eq. 9). This construction guarantees that the core interaction kernel is intrinsically **Symmetric and Positive Semi-Definite (PSD**). Consequently, the resulting Laplacian $\tilde{\mathbf{L}}(t)$ is symmetric. Regarding the PSD condition for $\tilde{\mathbf{L}}(t)$, it is true if the maximum eigenvalue of the normalized adjacency term is bounded by 1.
>
> **Q5. Initialization**
>
> We clarify that the initial latent state is initialized as a zero vector (i.e., $\mathbf{S}(t_{1}) = \mathbf{0}$).

---

### Official Review · Reviewer_CKa4 · 2025-10-31

**Soundness:** 3
**Presentation:** 3
**Contribution:** 3
**Rating:** 6
**Confidence:** 3

**Summary:**

TD-HNODE is a neat and technically sound synthesis -- temporally detailed hypergraph structure inside a neural ODE which is complemented with empirical improvements and ablations that attribute gains to both the attention based incidence and learnable trajectory weights.

Overall, I find the paper easy to read, well-written, and reasonably reproducible with the code provided as supplementary material. Further, it addresses a practical and relevant clinical modelling gap. The main caveats are clinical validity of the "irreversible" and pathway assumptions, narrow evaluation metrics for deployment, and baseline coverage -- which I would like the reviewers to clarify further.

**Strengths:**

Overall, the paper is well-motivated and tackles a problem of high relevance; namely addressing continuous-time disease progression with irregular visits and aligns modeling with clinically recognized pathways.

Summarising the positives below,

- Methodological novelty with clear mechanics. TD-HNODE combines a Neural ODE with a temporally detailed hypergraph: an attention-based, time-aware incidence matrix and a learnable hyperedge weight matrix to form a TD-Hypergraph Laplacian that drives ODE dynamics. This is a neat way to infuse high-order, pathway-level structure into continious modelling.
- Good experimentation including two real-world EHR datasets with patient-wise splits; additional cardiovascular disease experiments suggest some generality beyond diabetes which could further enhance the method applicability.
- Clear gains vs strong baselines + informative ablations. TD-HNODE tops Accuracy/Recall/F1 across T-LSTM, ContiFormer, TGNE, etc.
- Code is supplied in the supplementary material which greatly enhances reproducibility and dissemination of the work.

**Weaknesses:**

On the weaknesses for the paper can be summarised below,

- Labeling/trajectory assumptions may be clinically brittle. All markers are treated as irreversible first-occurrence events; including lab states like "HbA1c High/Low", "Poor Lipid/BP". This monotonicity simplifies modeling but risks mischaracterizing fluctuating conditions and inflating cumulative positives which can inflate results.
- Metric story is narrow for a clinical setting. Results emphasize Recall/Accuracy and macro F1; there’s no AUROC/AUPRC, calibration, decision-curve or time-to-event analysis. This is rather important in this setting as there are class imbalances and early-warning goals. Overall precision remains relatively low (e.g., 31.8% on MIMIC-IV), so false-positive indidence remains unclear.
- Baseline customization may under-serve competitors. Graph baselines are forced into pairwise graphs (breaking hyperedges), while discrete-time hypergraph baselines are snapshot-based. That’s fair to compare paradigms, but the paper should also include at least one continuous-time hypergraph variant.

**Questions:**

The questions that I have for the authors are described in the weaknesses of the paper that I mentioned, further I would like the authors to clarify the following,

- Tables report mean/std but not CIs/significance tests, and there’s no cross-site validation - why is that?
- There is also no analysis of shift when varying hypergraph definitions, and limited reporting on class prevalence which is key for clinical claims?

---

> ### Author Response · Authors · 2025-11-25
> **Response to Reviewer CKa4 (1/2)**
>
> We sincerely thank the reviewer CKa4 for the detailed feedback and for recognizing the significance of the problem. We have addressed the concerns raised point by point. **(W = Weakness, Q = Question)**.
>
> **W1. Labeling Assumptions**
>
> This is a great comment. We want to clarify that by "irreversibility", we mean that the disease progression stage can either stay unchanged or advance further, but does not revert to an earlier stage. Regarding states like "HbA1c High/Low", "Poor Lipid/BP", we actually preprocessed these states and used their first occurrence as a progression signal in our complication marker vector ($\mathbf{y}$). For example, once a patient has "Poor Lipid/BP", it indicates the chronic stage of a patient has changed to a new level that the patient **has ever experienced "Poor Lipid/BP"**. This progress of stage is irreversible. This design choice has been confirmed by a clinician (M.D.) in our co-author list. However, **our model also incorporates fluctuating lab states (e.g., GFR, HDL, Triglycerides)** of a patient using their raw values in the input risk factor features ($\mathbf{x}$).
>
>
> **W2. Metric Story**
>
> - **More Metrics**: Thanks for the suggestion! We have computed AUROC, AUPRC as shown in the table below, and we will include more evaluation metrics such as calibration, decision-curve or time-to-event analysis in the final revision.
>
> |             | University Hospital |       | MIMIC-IV |       |
> |-------------|---------------------|-------|----------|-------|
> | Methods     | AUROC               | AUPRC | AUROC    | AUPRC |
> | T-LSTM      | 0.625               | 0.046 | 0.616    | 0.058 |
> | ContiFormer | 0.833               | 0.108 | 0.802    | 0.100 |
> | MegaCRN     | 0.795               | 0.093 | 0.711    | 0.093 |
> | TGNE        | 0.840               | 0.117 | 0.775    | 0.101 |
> | DHSL        | 0.761               | 0.093 | 0.726    | 0.095 |
> | HyperTime   | 0.755               | 0.094 | 0.742    | 0.094 |
> | NODE        | 0.792               | 0.105 | 0.711    | 0.090 |
> | CODE-RNN    | 0.837               | 0.115 | 0.787    | 0.097 |
> | **TD-HNODE**    | **0.865**       | **0.142** | **0.816**    | **0.113** |
>
> The AUROC and AUPRC values of TD-HNODE indicate that TD-HNODE has better prediction performance than baselines. AUPRC is low across all methods mainly due to the extreme class imbalance—progression signals (class 1) are far sparser than class 0 (only around 2%), as detailed in our response to **Q2** below.
>
> Our clinician collaborator has advised that Recall (Sensitivity) is more important in progression modeling (screening for future complications). For example, **missing a progression event (e.g., Stroke) has far more severe consequences than a false alarm.**
>
>
> **W3. Baseline Customization**
>
> Thanks for this suggestion. We have added a continuous-time hypergraph baseline **TD-HNDOE_vanilla** by removing $\mathbf{H}_p$ and $\mathbf{W}_p$ from our model while keeping the same Neural ODE over the TD-hypergraph. As shown in the new table below, full TD-HNDOE consistently outperforms both HOPE and TD-HNDOE_vanilla on University Hospital and MIMIC-IV, validating the benefit of our temporally detailed TD-hypergraph dynamics.
>
> |                  | University Hospital |           |        |          | MIMIC-IV |           |        |          |
> |------------------|---------------------|-----------|--------|----------|----------|-----------|--------|----------|
> | Methods          | Accuracy            | Precision | Recall | F1-score | Accuracy | Precision | Recall | F1-score
> | TD-HNODE_vanilla | 0.731               | 0.106     | 0.761  | 0.155    | 0.830    | 0.233     | 0.731  | 0.308    |
> | TD-HNODE         | 0.794               | 0.143     | 0.793  | 0.204    | 0.879    | 0.318     | 0.857  | 0.429    |

---

> ### Author Response · Authors · 2025-11-25
> **Response to Reviewer CKa4 (2/2)**
>
> **Q1. Statistics**
>
> - **Significance**: We currently report mean $\pm$ std over multiple random runs for all methods. We will add confidence intervals / paired significance tests in the revision as suggested.
>
> - **Cross-site Validation**: We agree that cross-site validation is important. Unfortunately, we only have two de-identified real-world EHR data, and their feature spaces are not fully matched, so it is difficult to run cross-site validation. We will work with clinical collaborator to collect data from a different hospital system for cross-site validation in the future work.
>
> **Q2. Hypergraph Definitions & Class Prevalence**
>
> - **Hypergraph Definitions**: Thanks for the comment. Our hypergraph structure is constructed based on specific clinical pathways suggested by our clinical co-author (M.D.). We will add an analysis of shift when varying hypergraph definitions (e.g., perturbing pathway definitions).
>
> - **Class Prevalence**: We added a detailed table reporting the **positive prevalence (%) for each of the 21 complication markers** on both datasets (shown below). The average prevalence is extremely low (≈ 0.02), confirming severe class imbalance.
>
> |    Complication Markers   | University Hospital |  MIMIC-IV |
> |:-------------------------:|:-------------------------------------:|:--------------------------:|
> |         HbA1c_low         |                 0.058                 |            0.028           |
> |         Poor Lipid        |                 0.026                 |            0.039           |
> |        Hypertension       |                 0.068                 |            0.043           |
> |          Obesity          |                 0.058                 |            0.022           |
> |         Foot_ulcer        |                 0.005                 |            0.001           |
> | Blindness_and_vision_loss |                 0.003                 |            0.001           |
> |     Visual_impairment     |                 0.001                 |            0.001           |
> |  Congestive_heart_failure |                 0.018                 |            0.024           |
> |        Nephropathy        |                 0.030                 |            0.026           |
> |         Neuropathy        |                 0.051                 |            0.018           |
> |        Retinopathy        |                 0.011                 |            0.006           |
> |  Cerebrovascular_Disease  |                 0.011                 |            0.002           |
> |           Stroke          |                 0.011                 |            0.003           |
> |         Depression        |                 0.026                 |            0.038           |
> |        Hypoglycemia       |                 0.022                 |            0.003           |
> |         HbA1c_high        |                 0.053                 |            0.018           |
> |          Poor BP          |                 0.024                 |            0.005           |
> | Cardiac_revascularization |                 0.009                 |            0.010           |
> |    Atrial_fibrillation    |                 0.007                 |            0.021           |
> |           Cancer          |                 0.014                 |            0.001           |
> |        Ketoacidosis       |                 0.002                 |            0.019           |
> |       **Average**         |           **0.024**                 |         **0.016**           |
>
>
> Thanks again for these valuable comments from clinical perspective. We believe these additions will further strengthen the paper’s clinical clarity and empirical completeness.

---

### Official Review · Reviewer_rGrB · 2025-11-01

**Soundness:** 3
**Presentation:** 2
**Contribution:** 4
**Rating:** 6
**Confidence:** 4

**Summary:**

This paper addresses the prediction of disease progression from patient encounter data, incorporating risk factors such as medications, laboratory test results, and vital signs. The authors propose TD-HNODE, which combines a neural ODE and a hypergraph neural network, where Learnable TD-Hypergraph Laplacian plays a key role to make the model more data-driven and adaptable to patient-specific disease trajectories while maintaining its clinically verified nature. The proposed method was evaluated on two real-world EHR datasets and consistently outperformed baselines.

**Strengths:**

-- Learnable TD-Hypergraph Laplacian is a reasonable enhancement for the combination of neural ODE and hypergraph neural network, where Attention-based Indicence Matrix adjusts the degree of attributions of v in e flexibly according to the context, and Learnable Hyperedge Weight Matrix captures data-driven similarities between trajectories.

-- Experimental results on multiple datasets demonstrated the effectiveness of the proposed method.

-- The case study is interesting and practically important.

**Weaknesses:**

-- Clarity issues:
* In l.100, what is k?
* In l.147, The authors mentioned "we use the terms ‘hyperedge’, ‘pathway’, and ‘trajectory’ interchangeably", but this makes descriptions confusing. For example, p_j and e_j look the same, so we should use only one of them consistently throughout the paper.
* In Eq.2, LHS is better to be f(t,S(t),x(t);\Theta) not dS(t)/dt.
* In Eq.3, I is not defined.
* In the descriptions starting from l.242, e is used like an index of edge, but it was j until then. The index for the edge should be j, and the edge itself is denoted as e for consistency.
* In l.265, the temporally detailed hyperedge should have u on superscript.
* In l.269, e may not be the index for O and F, maybe. For indexing, only j is enough.

**Questions:**

Nothing.

---

> ### Author Response · Authors · 2025-11-25
> **Response to Reviewer rGrB**
>
> We sincerely thank the reviewer rGrB for the excellent rating on our contribution. We have incorporated all your suggestions into the revised manuscript to improve clarity. **(W = Weakness**).
>
> **W1. Clarity Issues**
>
> - $k$ denotes the index of the $k$-th encounter (timestamp) in a patient's sequence.
>
> - We apologize for the confusion. Conceptually, $p_j$ denotes the **ordered** sequence of disease progression, while $e_j$ represents the corresponding **node set** within the hyperedge. We agree with the reviewer that we should use only one of them consistently throughout the paper. This can be readily resolved in the revision.
>
> - Good suggestion! We have revised the LHS to $f\bigl(\,t,\mathbf{S}(t),\mathbf{x}(t);\boldsymbol{\Theta}\bigr)$ to align with the function definition.
>
> - We now explicitly define $\mathbf{I}$ as the **identity matrix** in the text following Eq. 3.
>
> - Thanks for spotting this. We will fix the notation and use $j$ as the hyperedge index consistently.
>
> - We earlier omitted patient index $u$ for simplicity (line l.191). We agree that adding back the index $u$ is less confusing.
>
> - Great suggestion! We have revised the index notation.

---

### Official Review · Reviewer_WqcR · 2025-11-01

**Soundness:** 2
**Presentation:** 3
**Contribution:** 3
**Rating:** 4
**Confidence:** 4

**Summary:**

In this work, the authors propose TD-HNODE that models disease progression along clinically recognized trajectories by constructing a temporally detailed hypergraph and capturing continuous-time progression dynamics through a neural ODE framework.

**Strengths:**

1. This provides a novel modeling method for EHR.

2. The figure is clearly diagrammed, and the notation table enhances the readability.

3. Both open-source and closed-source datasets are evaluated, validating its practice.

**Weaknesses:**

1. It is limited to evaluating the proposed method on only one category of EHR, i.e., type 2 diabetes. Other medical scenarios, e.g., Alzheimer’s disease, Parkinson’s disease, and chronic kidney disease (CKD), mentioned by the authors, are ignored.

2. The evaluation lacks soundness. Firstly, it is encouraged to involve the doctors' diagnosis by comparison. Secondly, the ODE steps need to extend to evaluate the robustness of TD-HNODE.

3. The writing is quite informal, e.g., we use the terms ‘hyperedge’, ‘pathway’, and ‘trajectory’ interchangeably.

4. The crucial baseline that models graph ODE, is ignored, e.g., HOPE[1].

5. The induction of equation 11 is absent. Why can an ODE capture this dynamic Laplacian matrix?

6. The related work is not explicitly illustrated.

[1] Luo, Xiao, et al. "Hope: High-order graph ode for modeling interacting dynamics." International conference on machine learning. PMLR, 2023.

**Questions:**

See the weakness.

---

> ### Author Response · Authors · 2025-11-25
> **Response to Reviewer WqcR**
>
> We sincerely thank the reviewer WqcR for the constructive feedback. Below, we address all concerns point by point. **(W = Weakness**).
>
> **W1. Additional Chronic Disease**
>
> We agree with the reviewer. In fact, we have already included additional experiments on a second disease category (cardiovascular disease), and it is in **Appendix D.7 of our original submission**. As summarized in Table 8, TD-HNODE consistently outperforms baselines on this dataset (e.g., Recall: 0.818 vs. 0.801 for TGNE). If preferred, we can highlight these results in the revised main body to demonstrate the model's generalizability beyond type-2 diabetes.
>
> **W2. Evaluation Soundness**
>
> - **Doctor's Diagnosis**: In fact, the diagnosis codes for disease complication biomarkers $\mathbf{y}_u(t)$ in our EHR data are collected in real-world clinical practice in hospitals. The codes are indeed from **clinicians (doctors)**.
>
> - **ODE Steps Robustness**:  Figure 3(b) in the paper presents a **sensitivity analysis** of TD-HNODE with respect to the number of ODE solver steps (ranging from 4 to 12). We further extend this analysis to 16 steps. The results show that performance improves as the number of steps increases and stabilizes around 10 steps.
>
> | Dataset             | 4     | 6     | 8     | 10    | 12    | 14    | 16    |
> |---------------------|-------|-------|-------|-------|-------|-------|-------|
> | University Hospital | 0.721 | 0.752 | 0.780 | 0.793 | 0.794 | 0.796 | 0.796 |
> | MIMIC-IV            | 0.743 | 0.770 | 0.844 | 0.857 | 0.859 | 0.861 | 0.860 |
>
>
> **W3. Terminology**
>
> Thanks for the suggestion. We will synchronize the terminology across the entire paper.
>
> **W4. Additional Baseline**
>
> Thank you for bringing up HOPE [1]. We have now included HOPE as an additional baseline below and will include these results in the revised paper. TD-HNODE consistently outperforms HOPE on both datasets:
>
>
> |          | University Hospital |           |        |          | MIMIC-IV |           |        |          |
> |----------|---------------------|-----------|--------|----------|----------|-----------|--------|----------|
> | Methods  | Accuracy            | Precision | Recall | F1-score | Accuracy | Precision | Recall | F1-score |
> | HOPE     | 0.725               | 0.090     | 0.724  | 0.161    | 0.829    | 0.229     | 0.715  | 0.299    |
> | TD-HNODE | 0.794               | 0.143     | 0.793  | 0.204    | 0.879    | 0.318     | 0.857  | 0.429    |
>
> These results show that our proposed TD-HNODE outperformed the baseline HOPE. The reason is likely that modeling clinical pathways using hypergraphs is better than using standard graphs, as a hypergraph captures **high-order** interactions between progression markers within a pathway.
>
> [1] Luo, Xiao, et al. "Hope: High-order graph ode for modeling interacting dynamics." International conference on machine learning. PMLR, 2023.
>
> **W5. Induction of Eq. 11**
>
> This is a very good question! Eq. 11 is inducted from the hypergraph neural ODE in Eq. 2 (line 215).
>
> Specifically, in Eq. 2, we construct a neural ODE with a hypergraph neural network, which learns the continuous-time patient progression state by incorporating the high-order interactions of progression markers along trajectories (hypergraph).
>
> However, Eq. 2 is based on a static hypergraph Laplacian, which does not encode how many complication markers a patient has **ALREADY observed** by time $t$ and how fast those markers occur. Thus, we extend Eq. 2 to Eq. 11 by replacing the static hypergraph Laplacian $\mathbf{L}$ with dynamic hypergraph Laplacian $\tilde{\mathbf{L}}(t)$. Here is the intuition. When a patient has an encounter at time $t$, additional disease-progression markers may be observed in the patient along a trajectory. The timestamps of these observed markers so far are explicitly encoded into $\tilde{\mathbf{L}}(t)$ via the continuous-time position encodings in Eq. 4 and 5 in an attention framework.
>
> We use the notation $\tilde{\mathbf{L}}(t)$ (rather than a static $\mathbf{L}$) to indicate that the Laplacian must be recomputed whenever a new encounter arrives at an irregular timestamp $t$.
>
>
> **W6. Related Work**
>
> We currently summarize the “related work” within the introduction. We can explicitly present these related works in a separate section of “Related Work”. We will also cite the baseline **Graph ODEs (HOPE)** as suggested.

---

### Meta-Review · Area_Chair_MjDs · 2026-01-11

**Summary:**

The main concerns are:
- Clinical assumptions, such as irreversibility of markers and pathway definitions.
- Evaluation completeness (missing metrics and baseline coverage).
- Deeper interpretability analyses, statistics, and robustness checks.
- Clarity, notation consistency and exposition.

**Reviewer Concerns:**

Concerns addressed:
- Missing baselines are addressed by adding HOPE and a continuous-time hypergraph ODE baseline.
- Evaluation completeness were improved by adding metrics such as AUROC and AUPRC; more analysis of ODE step sensitivity; clarifying clinician involvement; and justifying recall-focused evaluation under extreme class imbalance.
- Presentation improved.
- Generality beyond was evidenced by new experiments on cardiovascular disease.

Concerns remains:
- Depth of clinical validation.
- Statistical significance and cross-site validation.

**Reviewer Scores:**

It is likely that all reviewers except for wNfJ will have their score stay above acceptance threshold.

---

### Decision · Program_Chairs · 2026-01-26

Accept (Poster)